# Strategic Entrepreneurship and the Performance of Women-Owned Fish Processing Units in Cibinong District, Bogor Regency

Aditya Ari Yudhanto *, Emma Rochima and Rivani

Regional Innovation Graduate School, Universitas Padjadjaran, Bandung 45363, Indonesia
* Correspondence: aditya21014@mail.unpad.ac.id

**Abstract:** Strategic entrepreneurship refers to the ability of an MSME to investigate potential entrepreneurial ventures while exploiting its current competitive advantages. Academics and practitioners have offered models to deconstruct strategic entrepreneurship; however, there are few distinctive strategic entrepreneurship models appropriate for certain business circumstances. Culinary businesses in Cibinong District, Bogor Regency face several challenges, including low-quality human resources, inadequate capital and technology, and poor entrepreneurial spirit. This study aims to learn how the performance of women-owned fish processing MSMEs under COVID-19 conditions connects to several strategic entrepreneurship components, such as environmental factors, individual resources, resource orchestration, and competitive advantage. Research data taken from 30 women-owned fish processing businesses were processed using SMART-PLS 3.0, followed by a quantitative descriptive method analysis. The outcome was that the components of the environment, specific resources, and orchestration of those resources could generate performance and value for the customer, leading to competitive advantages. This research provides a current understanding of attitudes to businesswomen's activities throughout the pandemic period, particularly in relation to entrepreneurship chances and MSME performance. Strategic entrepreneurship is necessary to improve performance in dynamic environments.

**Keywords:** businesswomen; environmental factors; individual resources; organizational resources; resource orchestration; creating performance; entrepreneurship

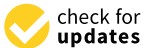



## 1. Introduction

The adoption of financial and nonfinancial initiatives within businesswomen's organizations has a number of benefits for businesses. Financial indicators are typically used to assess a company's efficiency; on the other hand, some nonfinancial measurements, such as customer loyalty and employee happiness, need to be considered and cannot be disregarded (Visedsun and Terdpaopong 2021). It is important to take into account how the organizational climate, including leadership, culture, and organizational structure, might impact an organization's success (Odongo et al. 2019).

One of the dynamic environmental factors that micro, small, and medium enterprises (MSMEs) must address in 2020 is the COVID-19 outbreak. Regrettably, social distance restrictions have decreased the number of customers, especially in the food and culinary industries. Many firms can prosper in a changing environment by seeking new business opportunities, such as incorporating activity on websites, applications, social media, e-commerce, and the exploitation of other resources. Despite the COVID-19 outbreak having an influence on the MSME food or culinary processing business, it is still rated as a high performer and a possible winner (Dcode Economic and Financial Consulting 2020). Entrepreneurs need to be more creative, aggressive, and competitive to survive and perform well in a dynamic economy. Strategic entrepreneurship is the term for this style of conduct (Ireland et al. 2003).



The concept of strategic entrepreneurship helps businesses, despite size and age, produce superior performance and preserve profits through opportunity- and advantage-seeking activities to reach prosperity in a dynamic and globalizing environment (Tülüce and Yurtkur 2015; Zucchella and Magnani 2016). Strategic entrepreneurship may increase the variety of goods on the market, create new market niches, and stimulate novel forms of competition (Kantur 2016). To increase profitability and market share, businesses of all sizes should integrate strategic entrepreneurship into their operations (Dogan 2015). MSMEs may be capable of continuing to achieve their primary goal of promoting economic growth, value creation, competitiveness, and employment (Awang et al. 2015).

A strategic entrepreneurial approach is appropriate for small and large organizations (Papulova and Papulova 2015). Entrepreneurship can influence national economic growth, including in Indonesia. Indonesia must quicken and elevate the caliber of its economic growth as a growing country. To create competitive entrepreneurial circumstances, it is essential to construct an entrepreneurial ecosystem (Iqbal et al. 2021). According to the Ministry of Cooperatives and MSMEs of Indonesia, MSMEs accounted for 97% of employment in 2021; moreover, they made up 61.97% of the country's GDP (Mariana 2022). MSMEs in the marine and fishery sectors have great potential for Indonesian business development. There are 60,855 small, medium, and large fish processing facilities in Indonesia (Directorate General of Competitiveness of Marine and Fishery Product Development of Marine dan Fishery Ministry of Indonesia 2019). Some of these are in Bogor Regency.

Bogor Regency is a second-level administrative region of West Java Province, Indonesia, and is very important as one of the buffer zones for the capital city. According to the Statistical Agency of West Java Province, 6,088,233 people lived in Bogor Regency in 2020, representing 12.19% of the entire population of West Java Province (Statistics of West Java Province 2021). Undoubtedly, there are significant opportunities and challenges for the economic growth of this population. One of the Bogor Regency's economic drivers is the expansion of micro, small, and medium-sized enterprises (Dewi 2020). Micro, small, and medium-sized businesses are essential for boosting the economy and creating jobs in many developing countries (Iqbal et al. 2021). Thus, the development of MSMEs in Bogor Regency could create job opportunities and enhance its economic growth.

The growth of micro, small, and medium-sized businesses, particularly seafood processing, is one of the factors driving the Bogor Regency economy (Dewi 2020). Micro and small firms in Bogor Regency face several challenges to their development, including low-quality human resources, inadequate capital and technology, and poor entrepreneurial spirit. Due to a number of factors, MSMEs typically have low levels of competitiveness, which can lead to business failure (Rainanto 2019). Fortunately, the existence of these businesses could be supported by enough resources. Bogor Regency produced over 115 thousand tons of 10 different varieties of consumable fish in 2020 (Statistics of Bogor Regency 2021). These fish have great potential to be turned into competitive, high-quality fishery products. According to the requirements of the Minister of Maritime Affairs and Fisheries Regulation Number 59 of 2021 concerning "The Increasing the Added Value of Fishery Products", the increase in added value is obtained by processing fresh fish into processed fishery products.

In contrast to other sub-districts, Cibinong District serves as the administrative hub of the Bogor Regency Government and is distinguished by the absence of a village government system throughout its whole administrative region. Cibinong District can be referred to as Cibinong City or can be considered an urban region. The majority of the people in this district have access to higher-quality infrastructure facilities, infrastructure of acceptable quality, and a strong network of banking literacy (Utami 2014). Business actors exist and grow in Cibinong District because of these benefits, particularly in the processing of fishery products. Of the 135 value-added fishery business players in the Bogor Regency, 30 were in Cibinong District. Additionally, it is interesting to note that all of them are run by women.

In addition to the benefits listed above, Cibinong District has excellent potential as a market for culinary goods made from processed fish, given its position as the administrative hub of Bogor Regency. Among the 39 districts in Bogor Regency, Cibinong District has the most commercial and service facility establishments (Machmud et al. 2021). Cibinong District also has the highest scalogram in the Bogor Regency's score for economic growth facilities, and is the hub of the economic boom in Bogor Regency. Cibinong District's labor structure has evolved, shifting from a predominance in the agricultural sector to the manufacturing and service industries (Utami 2014).

In a press release in 2021, the Ministry of Women's Empowerment and Child Protection of Indonesia mandated a message for women. They stated that in the COVID-19 pandemic era, the role of women in the family was directed to further increase their active participation in various development activities (Legal Bureau of the Ministry of Women's Empowerment and Child Protection of Indonesia 2021). This demonstrates that women can actively participate in socioeconomic activities, while at the same time playing a role as housewives or teachers to instill values in their children. One measure for empowering and improving the community's economy following the COVID-19 pandemic is to increase the role of women-owned fish processing MSMEs in Bogor Regency by utilizing various local potentials, referring to targets of the Sustainable Development Goals (SDGs) agenda, number five (Gender Equality) and number eight (Decent Work and Economic Growth).

Hence, despite encountering a variety of obstacles, MSMEs may continue to fulfill their primary responsibility of fostering economic growth, value creation, competitiveness, and employment. According to the definition above, a strategic plan needs to be developed for the sustainability of women-owned fish processing MSMEs. Strategic entrepreneurship is a planning and forecasting technique used to take full advantage of possibilities when competing and outperforming rivals.

Strategic entrepreneurship is one way to gain competitive advantage supported by creativity and innovation (Ireland et al. 2003). The ability of business actors to manage their resources with the help of leadership, culture, and an entrepreneurial attitude is an example of this type of innovation. These three things are the core of entrepreneurship. Strategic entrepreneurship research is still conceptual and not based on empirical findings (Ireland et al. 2003).

Based on this, researchers have the chance to conduct empirical research by identifying the traits of women who process fishery products, and by examining the relationships between variables in the strategic entrepreneurship model that are related to the success of MSME women who process fishery products in Bogor Regency. Women who process fishery products will be identified based on their age, level of education, business ownership status, business history, time of business establishment, sources of business capital, total assets, total turnover, and number of employees.

It is obvious that strategic planning must be devised to secure the long-term viability of MSMEs. An approach to making the most of the prospects in Cibinong District, in terms of addressing retail rivalry and boosting the sector's competitiveness, is through strategic entrepreneurship. This study investigates the characteristics of women-owned fish processing units in Bogor Regency, strategic entrepreneurial considerations, and the influence of input-process-output segmentation to maximize opportunities by fostering competition in MSME business processes.

The subject of this study is fish processing MSMEs in Cibinong District owned by women. The purpose of this study is to comprehend the relationship between the performance of women-owned fish processing MSMEs under the conditions of COVID-19 and the components of strategic entrepreneurship. The findings of this study will be useful for the development of governmental intervention strategies that suit the needs of businesswomen involved in the processing of fish. This study also applies structural equation modeling using partial least squares (PLS-SEM) to a new field.

## 2. Materials and Methods

This study used a survey method using a questionnaire, with women-owned fish processing units serving as the direct respondents, to provide a thorough account of the circumstances surrounding a case. Data collection, surveys, and the direct distribution of questionnaires to respondents were all part of the search methodology, which focused on women-owned fish processing facilities in Cibinong District, Bogor Regency.

According to the Statistics of West Java Province in 2020, the population of Bogor Regency was 6,088,233, or 12.19% of the total population of the province (Statistics of West Java Province 2021). Bogor Regency consists of 40 districts (Saraswati 2014), and is the most populated resident regency in Indonesia (Directorate General of Population Affairs and Civil Registration of the Ministry of Home Affairs of Indonesia 2021; Kusnandar 2021). Cibinong District is one of the most densely populated districts in Bogor Regency.

In contrast to other sub-districts, Cibinong District, which serves as the administrative hub of the Bogor Regency Government, is distinctive in that the entirety of its administrative region lacks a village government structure. The bulk of the population in this district benefits from better access to education, high-quality infrastructure, and a strong network of banking literacy (Aprilia et al. 2021; Utami 2014). Figure 1 provides a map of the area studied in Cibinong District, Bogor Regency.

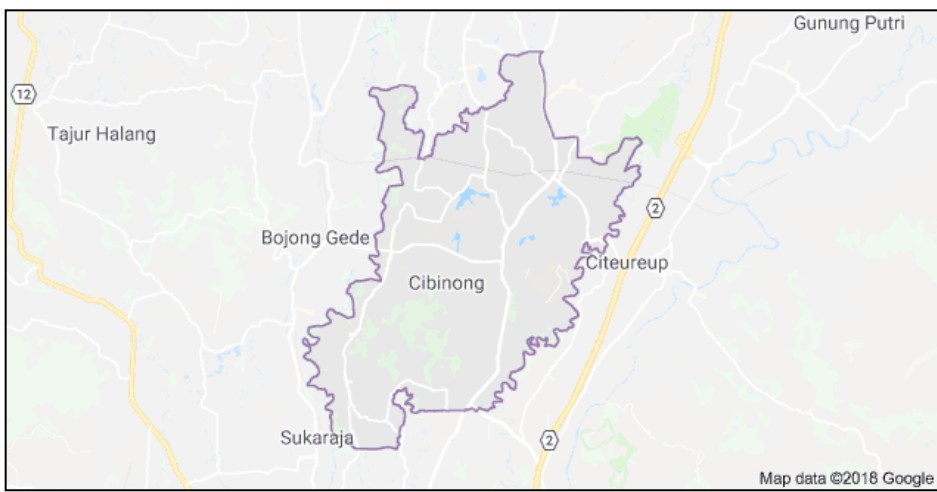

**Figure 1.** Cibinong District Map of Bogor Regency.

### 2.1. Census Data Collection Method

For this analysis, we combined primary and secondary data. Primary data were collected directly via a questionnaire to determine how effectively management understood strategic entrepreneurship. This study was carried out between August 2022 and October 2022 in Cibinong District, Bogor Regency with a census of 30 respondents of commercial businesswomen fish processing units. Secondary data on strategic entrepreneurship were gathered from a variety of relevant literary works, including journals, books, earlier study findings, and statistical data reports (Hair et al. 2017).

The goal of the questionnaire was to assess the six aspects of strategic entrepreneurship (environmental factors, organizational resources, individual resources, resource orchestration, creating value and advantage, and creating performance). The environmental factors were measured using items developed by Revilla et al. (2011) and Tang (2008), while the organizational resource items proposed by Hitt et al. (2011) were used. Resource orchestration was measured by adapting items from Carnes et al. (2017), creating value and advantage by adapting items from Porter (2007), and creating performance by adapting the items from Shepherd and Wiklund (2009). All items were evaluated using a Likert scale of one to five, with five expressing strong agreement. The Likert scale is used to convey how strongly respondents agree or disagree with specific statements about actions, things,

people, or events. The suggested scale typically consists of five points. A Likert scale was chosen with five class scores as the measurement. There are a total of five groups, made up of the average value of each informant. The following formula can be utilized to determine class intervals:

$$SR = (a - b)/c$$

Explanation:k
SR = Range.
a = Maximum scores.
b = Minimum scores.
c = Number of class intervals.

$$SR = (5 - 1)/5$$
$$= 0.8$$

These calculations enable us to establish that the calculated scale range is 0.8. According to the statement on the research questionnaire, the average range of 1.00–1.80 falls into the Poor category, >1.80–2.60 falls into the Fair category, >2.60–3.40 falls into the Good category, >3.40–4.20 falls into the Very Good category, and >4.20–5.00 falls into the Excellent category. The items used to measure each variable are listed in Table A1 in Appendix A.

The selection of micro and small-scale criteria refers to article 35 of Government Regulation number 7 of 2021 (Government of Indonesia 2021). This law states that micro standards have an annual revenue of fewer than 2 billion rupiahs and business capital of no more than 1 billion rupiahs, excluding land and structures. In addition, small-scale businesses are considered to be businesses that do not include land and buildings, with yearly sales of between 2 billion and 15 billion rupiahs and business capital of between 1 billion and 5 billion rupiahs.

*2.2. Data Analysis*

The data processing and analysis for this study employed partial least square structural equation modeling, validity testing, reliability tests, descriptive analyses, and Simulation of Partial Least Square Structural Equitation (PLS-SEM). In this study, Smart PLS 3.0 was used. Descriptive analysis was performed to obtain a wide description of the characteristics of respondents, including gender, age, education, and MSME profile, as well as to describe strategic entrepreneurship implementation using the mean. For the measurement, a 5-point Likert scale was used to determine the scale range. A Likert scale is used to assess responders' attitudes, views, and perceptions of social issues (Sugiyono 2017).

The measurement model (outer model) and the structural model (inner model) are the two sub-models that make up PLS-SEM analysis (Hair et al. 2014). The constructs' convergent validity, discriminant validity, and reliability are assessed using the outer model (Hair et al. 2017). In addition, the inner model is used to assess the relevance of the path coefficients and the R-square value. Two categories of variable are used in PLS-SEM. The first is an observed variable, sometimes known as a manifest variable because it can be seen immediately. The second category is unobserved variables, sometimes known as latent variables since they cannot be observed directly (Hair et al. 2014). Together with the seven latent variables, there are 36 manifest variables (environmental influences, organizational resources, individual resources, resource orchestration, competitive advantage and value creation, and performance creation).

This research mainly employed the strategic entrepreneurship model based on Hitt et al. (2011) combined with Kiyabo and Isaga (2019) model. This study adopted Hitt et al. (2011) input-process-output model, extending the understanding of the strategic entrepreneurship construct. This used environmental factors, organizational resources, and individual resources as inputs, along with resource orchestration as the process, and creating value for customer advantages as the outputs. Another output was added: SME performance, from Kiyabo and Isaga (2019) model. Hence, from these models, this study generates the

input, process, and output of the strategic entrepreneurship model with inputs such as environmental factors (X1), organizational resources (X2), and personal resources (X3). The process segment has a variable latent resource orchestration (X4). Thus, the output section contains two variables: producing value and competitive advantage (X5), and creating performance (Y). Figure 2 displays the research model.

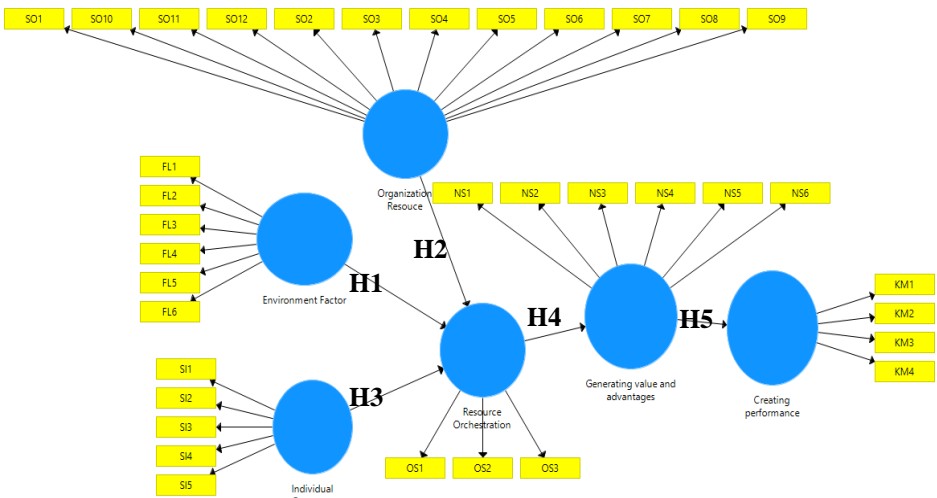

**Figure 2.** The Research Model.

The research hypotheses, as shown by the research model in Figure 2, are as follows:

**H1:** *Environmental factors (FL) have a positive effect on resource orchestration (OS).*

**H2:** *Organizational resources (SOs) have a positive effect on resource orchestration (OS).*

**H3:** *Individual resources (SI) have a positive effect on resource orchestration (OS).*

**H4:** *Resource orchestration (OS) has a positive effect on creating value and competitive advantage (NS).*

**H5:** *Creating value and competitive advantage (NS) has a positive effect on performance creation (KM).*

The outer model evaluation and the inner model evaluation are the two evaluation models utilized in PLS-SEM data analysis (Cheung 2013). Outer models are used to examine the effects of latent variable indicators. Multicollinearity was employed in this work to clarify the data without any discernible bias prior to analysis. The absence of a multicollinearity issue is a prerequisite for properly examining the outer model. A situation with substantial correlation or connectedness between indicators is called multicollinearity. A variance inflating factor (VIF) value of more than five indicates a multicollinearity correlation value, which is defined by a correlation value of more than nine. Multicollinearity is present if the latent variable VIF value is more than five. The actions that can be taken include lowering or eliminating indications with a high degree of association (Hair et al. 2017).

The evaluation of the outer model consists of three tests. A convergent validity test can be used to assess how well manifest variables can explain hidden variables by looking at loading factors above 0.50. When the average variance extracted (AVE) result is more than 0.50, the discriminant validity test is used to assess how many latent variables and manifest variables differ from one another. A previous study explained the connection between Cronbach's alpha above 0.60 and composite reliability used to test composite reliability (Hair et al. 2017). The inner model is utilized to determine the effect of the independent variable on the dependent variable by comparing the coefficient of determination (R square) and the path coefficient (Ghozali 2015).

## 3. Results and Discussion

### 3.1. Common Method Bias

A problem known as common method bias (CMB) occurs when the measuring technique utilized in an SEM study causes issues, rather than the network of causes and effects among latent variables in the model under investigation (Kock 2015). In this study, Smart PLS was used to identify CMB threats. The test signified that the VIF elements were lower than the 3.3 threshold. This indicates that the model is free from CMB (Hair et al. 2017; Kock 2015).

### 3.2. Model Measurement

The fit of the measurements was examined using validity and reliability standards. The ability of a measuring device (or objects) to consistently produce the same result is known as reliability. Validity is a measure of how accurately a notion is measured by a measuring tool (items). Since there are multicollinearity conditions, the actions that can be taken include lowering or eliminating indications with a high degree of association. The outcomes of VIF measurements at the manifest variable level for all latent variables in Table 1 are listed below, while a summary of the model's measurements after the multicollinearity test is shown in Table 2.

**Table 1.** Variable Manifest (VM) VIF Measurement Results.

| Items | KM | NS | FL | SI | SO | OS |
|---|---|---|---|---|---|---|
| KM | | | | | | |
| NS | 1.000 | | | | | |
| FL | | | | | | 1.857 |
| SI | | | | | | 3.092 |
| SO | | | | | | 3.225 |
| OS | | 1.000 | | | | |

Source: Compiled by the author.

**Table 2.** Summary of model measurements after multicollinearity test.

| Item | Indicator | Measurement Result | | | | | | Supported |
|---|---|---|---|---|---|---|---|---|
| Outer Loading | >0.7 | FL2 | 0.977 | SI3 | 0.940 | KM2 | 0.831 | Yes |
| | | FL6 | 0.967 | OS1 | 0.900 | KM3 | 0.871 | |
| | | SO4 | 0.747 | OS2 | 0.819 | KM4 | 0.816 | |
| | | SO5 | 0.829 | OS3 | 0.872 | | | |
| | | S09 | 0.754 | NS2 | 0.829 | | | |
| | | SO10 | 0.890 | NS4 | 0.904 | | | |
| | | SO12 | 0.869 | NS6 | 0.834 | | | |
| | | SI2 | 0.806 | KM1 | 0.745 | | | |
| Average Variance Extracted (AVE) | >0.5 | FL | | 0.944 | | OS | 0.747 | Yes |
| | | SO | | 0.672 | | NS | 0.734 | |
| | | SI | | 0.767 | | KM | 0.668 | |
| Composite Reliability | >0.6 | FL | | 0.971 | | OS | 0.899 | Yes |
| | | SO | | 0.911 | | NS | 0.892 | |
| | | SI | | 0.867 | | KM | 0.889 | |
| Cronbach Alpha | >0.6 | FL | | 0.941 | | OS | 0.830 | Yes |
| | | SO | | 0.877 | | NS | 0.823 | |
| | | SI | | 0.714 | | KM | 0.841 | |

Source: Compiled by the author.

The statements in the questionnaire were valid at a significance level of 5%, where r counts surpassed r tables based on the validity and reliability results of the 30 samples (0.361). In this study, each variable's Cronbach's alpha value was greater than 0.06, which indicates the dependability of the variables.

The Fornell–Larcker criterion, a gauge of the anticipated degree of "difference" between items for various factors, was used to test discriminant validity. The correlation square was compared to the AVE of each factor to assess the discriminant validity of the model. The other numbers are the correlation coefficients between the factors, which are thought to have excellent discriminant validity when the AVE is greater than the correlation coefficient between the factor and the other factors. The value on the diagonal represents the square root of the AVE (Hair et al. 2017). Values off the diagonal are correlations, whereas values (on the diagonal) represent the square root of the AVE. The discriminant validity results are shown in Table 3.

**Table 3.** Discriminant validity matrix.

| Items | KM | NS | FL | SI | SO | OS |
|-------|-------|-------|-------|-------|-------|-------|
| KM | 0.817 | | | | | |
| NS | 0.651 | 0.857 | | | | |
| FL | 0.372 | 0.809 | 0.972 | | | |
| SI | 0.583 | 0.773 | 0.636 | 0.876 | | |
| SO | 0.491 | 0.752 | 0.655 | 0.811 | 0.820 | |
| OS | 0.512 | 0.813 | 0.680 | 0.820 | 0.724 | 0.864 |

Source: Compiled by the author.

### 3.3. Respondent Characteristics

One of the respondents in this study was the owner of a fish processing company; thus, the sample consisted of one small business and 29 microbusinesses. Both characteristics are mentioned in Table 2. In addition, 50% of respondents in small businesses were between the ages of 41 and 50. This age group was regarded as still being capable and ready for the workplace. Most of them possessed either a bachelor's degree (37%) or senior high school degree (37%) as their last degree. Moreover, 87% of business units have been in operation for five to ten years. The respondent who owned the small business was 61 years old and had completed senior high school. In addition, the business had been in operation for more than 20 years. The respondent characteristics are shown in Table 4.

**Table 4.** Respondent Characteristics.

| Category | Subcategory | Unit | % |
|----------|-------------|------|---|
| Age (years old) | 21–30 | 4 | 13% |
| | 31–40 | 5 | 17% |
| | 41–50 | 15 | 50% |
| | >51 | 6 | 20% |
| Last Education | High School/Equal | 11 | 37% |
| | Diploma | 8 | 27% |
| | Bachelor | 11 | 37% |
| Ownership status | Own | 30 | 100% |
| Business Establishment Background | Own initiative | 29 | 97% |
| | Match to educational background | 1 | 3% |
| Age of organization (years) | <1 | | 0% |
| | >1–5 | 3 | 10% |
| | >5–10 | 26 | 87% |
| | >10–20 | 1 | 3% |
| Working capital | Own capital | 30 | 100% |
| Amount of Assets (Million Rupiahs) | >50–500 | 27 | 90% |
| | >500 | 2 | 6.67% |
| | >1000 | 1 | 3.33% |
| Revenue (Million Rupiah/month) | 25–75 | 29 | 97% |
| | 75–100 | | 0% |
| | >100 | 1 | 3% |
| Amount of manpower (person) | >3–10 | 29 | 97% |
| | >10 | 1 | 3% |

Source: Compiled by the author.

### 3.4. Descriptive Analysis of Strategic Entrepreneurship

Descriptive analysis was used to examine how entrepreneurs regarded the implementation of strategic entrepreneurship management in Bogor Regency's women-owned fish processing unit MSMEs based on three input variables: environmental factors, organizational resources, and personal resources. The process variables, namely resource orchestration and output variables, were composed of competitive advantage, value creation, performance generation, and benefits.

Three criteria—dynamics, munificence, and complexity—were used to assess environmental elements as external determinants. A good grade of 4.467 was used to estimate the variable's average value. This outcome demonstrated the superior environmental support provided by the MSME Business System for fish processing units in Bogor Regency. The least favorable average, which was nonetheless seen favorably, indicated capital support from banks and investors. Several business owners chose to borrow money privately instead of applying for bank loans because banks require guarantees, or have challenging application processes and onerous rules. Table 5 displays the descriptive analysis findings.

**Table 5.** Descriptive Analysis of Strategic Entrepreneurship.

| Latent Variable | Manifest Variable | Code | Mean Value | Category |
|---|---|---|---|---|
| Environment Factor | Consumer demand | FL1 | 4.633 | Excellent |
| | Business environment | FL2 | 4.467 | Excellent |
| | Capital access | FL3 | 4.233 | Excellent |
| | Fulfillment resource | FL4 | 4.567 | Excellent |
| | Production complexity | FL5 | 4.467 | Excellent |
| | Marketing complexity | FL6 | 4.433 | Excellent |
| Organizational resource | New ideas and creativity | SO1 | 4.233 | Excellent |
| | Risk taking | SO2 | 4.267 | Excellent |
| | Failure is tolerated | SO3 | 4.400 | Excellent |
| | Learning is promoted | SO4 | 4.433 | Excellent |
| | Supporting innovation | SO5 | 4.300 | Excellent |
| | Continuous change | SO6 | 4.600 | Excellent |
| | Commitment to improvement | SO7 | 4.633 | Excellent |
| | Protect innovation threatening | SO8 | 4.367 | Excellent |
| | Delivering opportunities | SO9 | 4.200 | Very Good |
| | Reasonable entrepreneurship | SO10 | 4.333 | Excellent |
| | Revisit entrepreneurship principle | SO11 | 4.467 | Excellent |
| | Link strategic management and entrepreneurship | SO12 | 4.533 | Excellent |
| Individual Resource | Introducing opportunity | SI1 | 4.533 | Excellent |
| | Entrepreneurial agility | SI2 | 4.333 | Excellent |
| | Real logical thinking | SI3 | 4.600 | Excellent |
| | Framework of entrepreneurship | SI4 | 4.467 | Excellent |
| | Opportunity collection | SI5 | 4.500 | Excellent |
| Resource Orchestration | Structuring resource portfolio | OS1 | 4.567 | Excellent |
| | Bundling resource | OS2 | 4.433 | Excellent |
| | Leverage capability | OS3 | 4.467 | Excellent |
| Creating value and competitive advantage | Customer relationship | NS1 | 4.567 | Excellent |
| | Different in service | NS2 | 4.467 | Excellent |
| | Business cost | NS3 | 4.500 | Excellent |
| | Differentiation | NS4 | 4.567 | Excellent |
| | Focus | NS5 | 4.700 | Excellent |
| Creating performance | Able to tax payment | KM1 | 4.367 | Excellent |
| | Improvement in asset | KM2 | 4.633 | Excellent |
| | Creating new jobs | KM3 | 4.533 | Excellent |
| | Improvement in selling | KM4 | 4.767 | Excellent |

Source: Compiled by the author.

A mean value of 4.397 for the organizational resource variable suggests that entrepreneurial leadership and entrepreneurial culture fall into the excellent category. Entrepreneurial leadership is the capacity to persuade others to pursue their objectives, search for possibilities, manage resources strategically, and foster an entrepreneurial environment that will help them stay competitive (Fontana and Musa 2017). Entrepreneurial culture is the term used to describe a company's culture that values not only seeking opportunities but also making money (Utoyo et al. 2020). On the other hand, the indicator of business owners providing opportunities had the lowest average value, indicating that MSMEs were still unable to adequately inform their employees.

The individual resource variable, which had an average of 4.487 responses, can demonstrate an entrepreneur's entrepreneurial attitude toward the opportunities directed to business development with the commitment, decisions, and actions to pursue the opportunities in dynamic environmental conditions (Gillin et al. 2019). This result gave the rating 'excellent'. On the other hand, the average for the entrepreneurial agility score was the lowest. It might be said that Cibinong District's fishery product processors continually adjust to market conditions in an effort to maintain the continuity of their businesses.

The capability to manage resources (structure), resources that turn into capabilities (bundling), and capabilities to provide value for customers were all included in the resource orchestration variable, which similarly had a mean value of 4.489 (leveraging). This placed it in the excellent category. Strategic entrepreneurship management demonstrated a great ability to give MSMEs a competitive edge, create value, and deliver performance and benefits for businesses, people, and society.

### 3.5. Analysis Results

With a loading factor value of 0.50 and no multicollinearity issues, 17 of the 36 indicators under examination pass the convergent validity test, according to the assessment outer model (Table 1). The study model was cleared of the indicators FL1, FL3, FL4, FL5, SO1, SO2, SO3, SO6, SO7, SO8, SO11, SO12, SI1, SI4, NS1, NS3, and NS5. In the discriminant validity test, every latent variable had an AVE value greater than 0.50. Figure 3 depicts the final model.

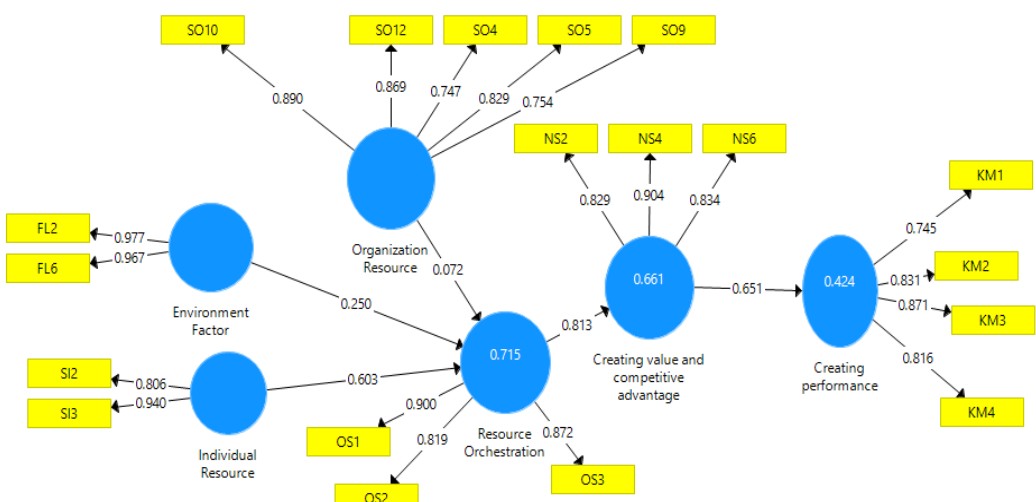

**Figure 3.** Final Model.

All the latent variables in the composite reliability test were known to have alpha Cronbach values of 0.60, and those variables matched the requirements for the composite reliability test. This follows the concept that the study model can be accepted as valid and credible by eliminating the eleven variables. The inner model was assessed by analyzing the R-square and path coefficient values. In addition, this model used the R-square value to determine how much an exogenous variable would influence an endogenous variable.

The test results showed that 71.5% of the variability in resource orchestration can be accounted for by the input segment's R-square value, including the variables for environmental conditions, organizational resources, and individual resources (Table 6). Sixty-six percent of the elements influencing value and competitive advantage in the process segment can be attributed to the resource orchestration variable. In the output sector, 42.4% of the creating performance variable may be attributed to both competitive advantage and creating value.

**Table 6.** Result of R-Square.

| Variable | R-Square |
|---|---|
| Resource orchestration | 0.715 |
| Creating value and competitive advantage | 0.661 |
| Creating performance | 0.424 |

Source: Compiled by the author.

The original sample value, *p* value, and t-statistic values were generated by the significance test of the route coefficient using the bootstrapping technique to examine the research model and research hypothesis. The initial sample values made it clear how the variables related to others. When the variable had a positive influence, it went from being negative to positive, and vice versa. The t-statistic value could be used to calculate the significance of a hypothesis test. The hypothesis was supported because the *p* value was less than 0.05. The route coefficient value, which shows the relationship between all variables, is shown in Table 7.

**Table 7.** Path Coefficients.

| Path | Original Sample | T-Statistic | *p*-Value | Hypotheses |
|---|---|---|---|---|
| FL → OS | 0.250 | 1.889 | 0.030 | H1 Accepted |
| SO → OS | 0.072 | 0.376 | 0.354 | H2 Not Accepted |
| SI → OS | 0.603 | 3.411 | 0.000 | H3 Accepted |
| OS → NS | 0.813 | 10.586 | 0.000 | H4 Accepted |
| NS → KM | 0.651 | 7.610 | 0.000 | H5 Accepted |

Source: Compiled by the author.

According to the test results, environmental factors have a considerable impact on orchestration resources since the *p* value is lower than 0.005, which is 0.030; therefore, hypothesis 1 is accepted. Organizational resources have no discernible impact on orchestration resources because the *p* value of 0.354, which is higher than 0.005, indicates that H2 cannot be accepted. Hence, individual resources have a favorable and considerable impact on orchestration resources. Moreover, the *p* value of H3 being lower than 0.005, which is 0.000, indicates that H3 is accepted.

Thus, it can be concluded that resource orchestration significantly affects value creation and competitive advantage. The *p* value of H4 is also less than 0.005, proving that H4 is accepted. The *p* value of H5 being substantially lower than 0.005 shows that H5 is accepted. Thus, providing value and gaining a competitive edge have a favorable impact on production performance.

The test results of this research are similar to those of previous empirical studies, and show that environmental conditions significantly affect resource orchestration in the input segment. The entrepreneur's local, physical, psychological, and social environments make up their environment (Grundstén 2004). Environmental circumstances might affect an entrepreneur's ability to compete and build company procedures in one of two ways: either as strength or weakness. It is necessary for entrepreneurs in the fish processing business sector to make efforts to recognize changes in the business environment if they are to maximize their ability to turn environmental influences into strengths.

Additionally, it is vital to comprehend the degree of complexity of marketing, service, and operational processes as well as the resources available to satisfy the needs of resources utilized

in fish processing business processes (Revilla et al. 2011). Based on the findings of the descriptive study, it is still difficult for women-owned fish processing firms in Cibinong District, Bogor Regency to discover specific information about how to obtain finance from banks or investors.

There are two key components in organizational resources: entrepreneurial attitude, and entrepreneurial culture (Ireland et al. 2003). The business owner or leader plays a particularly substantial part in preparing management and staff to implement a culture that fosters the growth of both employees' abilities and the application of ideas and innovations in retail (Utoyo et al. 2020). On the other hand, an entrepreneur can develop fresh resources that produce wealth, or endow existing resources to do so.

Employees will be motivated to adopt an entrepreneurial mindset, which is essential to the success of the company, not simply as a result of an entrepreneurial environment's ability to inspire innovation (Cho and Lee 2018). This will have an impact on how effectively and efficiently the fish processing unit's resources are managed. The results of the descriptive study, however, indicate that women-owned fish processing businesses in Cibinong District, Bogor Regency are still not doing the best job of giving their employees the opportunity to take chances with their responsibilities and authority at work.

The impact of the individual resource variable on resource orchestration was shown to be both favorable and significant. This demonstrates that women-owned fish processing businesses in Cibinong District, Bogor Regency, tend to be less than ideal at managing information about market development to find new opportunities and less than ideal at evaluating the benefits and drawbacks of proposed product innovations. According to the researchers, this is due to management being dominated by people between the ages of 41 and 50 who are high school graduates and whose experience and skills are insufficient, particularly in retail management connected to decision-making and strategic management.

According to the findings of the descriptive study, the women's business processing unit in Cibinong District, Bogor Regency, is still not doing the best job of managing information opportunities in the region's fish processing industry. Women-run fish processing businesses must focus on the cooperative operation of organizational and personal factors and the combination of individual and organizational elements.

Value creation and competitive advantage are significantly impacted by the resource orchestration variable. To determine the strategy used in the application of producing value and competitive advantage in retail, it is crucial to identify, create portfolios, group, improve, and enhance the competence of resources owned by retailers. Additionally, a combination of input groups will offer shops various tactics they can use. Based on the findings of the descriptive analysis, it can be concluded that the women-owned fish processing businesses in Bogor Regency have not given their resources the necessary attention. Therefore, it is essential to provide programs that assist staff in developing their knowledge and skills.

Performance creation is positively impacted by the variables providing value and competitive advantage. This demonstrates how efforts to improve product competitiveness and consumer value have a significant impact on performance creation. This is in line with the strategic entrepreneurship paradigm (Hitt et al. 2011). Effective strategic entrepreneurship would aid businesses in gaining an edge over rivals and adapting to environmental changes, which would benefit businesspeople, organizations, and societies overall, and have a considerable positive impact on economic performance (Awang et al. 2015).

### 3.6. Managerial Implications

The managerial ramifications of this study were used to enhance the strategic entrepreneurship management effectiveness of female MSMEs in fish processing units in Bogor Regency. Based on the lowest average descriptive analysis result, an analysis of the impact of strategic entrepreneurship implementation in input segments could be developed.

The most substantial impact on resource orchestration came from individual resources. For MSMEs, problems in the dynamic environment include fiercer and more widespread competition as well as client demands for the quality of various services and goods. There-

fore, entrepreneurs must develop a vision toward positive transformation and innovation. Thus, they must enhance their organizational capabilities if they want to survive and expand their business.

They should increase their organization's capacity for accepting, authorizing, and embracing constructive change. Therefore, it is important to modify how those involved in the transformation process perceive one another. Organizational reform is necessary for the benefit of both the organization and its personnel (worthiness).

Briefing and sharing sessions with stakeholders and staff could be used to implement this kind of change. By creating a good work environment that can foster employees' creativity and independence in submitting innovative ideas, they could establish an innovative culture to produce innovations in new goods and processes.

Entrepreneurs must enhance organizational capacities so that organizations may better utilize their assets, personnel, and operational procedures. To explore the competencies required and the resources that should be gathered or maintained, they must have a systematic framework. Human resources (HR) are crucial to the success of managing strategic entrepreneurship by enhancing organizational capabilities. To encourage business actors to be more competitive, they must upgrade their skills, particularly in HR management, by integrating HR into training programs, exhibition events, entrepreneurship seminars, and technical guidance (Dewi 2020).

## 4. Discussion

H2 was taken into consideration, and it was found that the $p > 0.05$ did not reach the level of significance. This finding led to the rejection of the premise that organizational resources significantly affect resource orchestration for women-owned fish processing businesses in Cibinong District, Bogor Regency. This suggests that strategic entrepreneurship management significantly affects MSMEs' capacity for innovation.

The research verified a significant relationship between resource orchestration, value creation, and competitive advantage for MSME performance. According to the research, businesses would have a better chance of enhancing the quality of their products if resource orchestration management effectively positioned its resource portfolio, improved human resources, and integrated the use of opportunity- and advantage-seeking behavior. This result is consistent with Porter Diamond's national advantage argument. The findings are in line with the theory advanced by Hitt et al. (2001), Hughes et al. (2021), Okoi et al. (2022) and Utoyo et al. (2020) that organizational performance has an impact on both direct and indirect resource management.

The total cost of investing in human capital is greater than the value of the results generated, or the cost of investing in human capital is greater than the results produced. The resource-based perspective also assumes that when MSME resources are objectively maximized, shareholder value will also be maximized. The findings support Okoi et al.'s (2022) claim that by nurturing a certain culture, knowledge, and abilities within an MSME, human capital management strategies inside the organization will play a crucial role in the maintenance of competitive advantage over competitors.

Additionally, the Porter Diamond Model of National Advantage is connected to the result. This result disproves the notion that innovative entrepreneurship has no measurable effect on MSME performance. This implies that entrepreneurship innovation has a major impact on the performance of MSMEs. The study provided evidence that there is a significant connection between entrepreneurial value and competitive advantage and MSME performance (Latianingsih et al. 2022; Okoi et al. 2022).

According to the findings, if new ideas are fostered and a paradigm shift is noticed, the organization's sales volume and profits will experience a boost, elevating the performance of the MSME. The results are consistent with the theory that entrepreneurial innovation is the readiness to encourage imagination and experimentation within the firm, as well as the utilization of technological leadership and R&D. Customers become aware of the worth of the goods or services given because of the innovation (Lumpkin and Dess 2001).

The performance of Cibinong's women-owned fish processing MSMEs during the pandemic period and the entrepreneurial component of strategic entrepreneurship are positively associated. During such periods, entrepreneurs face considerable risks that affect their financial situation and, ultimately, their capacity to survive (Kunc and Bandahari 2011; Pal et al. 2014). However, significant economic shocks can encourage the adoption of new business models and technologies (Brodherson et al. 2017) and present new opportunities (Beliaeva et al. 2020; Hausman and Johnston 2014; Pearce and Michael 2006). Therefore, new products, services, and business models are tested by women owners of fish processing MSMEs in Cibinong, which tend to be less affected by economic downturns.

According to studies using data from developed and emerging economies, rising economic pressure frequently encourages businesses to adopt innovative choices that have a favorable impact on their financial performance (Beliaeva et al. 2020; Hausman and Johnston 2014). Innovative companies also improve their market share and dominance (Pearce and Michael 2006; Guellec and Wunsch-Vincent 2009). Therefore, business decisions are crucial in times of crisis and become important success elements for MSMEs (Sahut and Peris-Ortiz 2014). In contrast, neither the performance of businesses nor the strategic element of SE, nor the industry-specific actions taken by Cibinong's MSME businesswomen during the COVID-19 epidemic, were found to be statistically associated.

The proposed strategic concept of entrepreneurship within the context of strategic management theory, with a focus on individual strategic entrepreneurship components (environmental factors, individual resources, resource orchestration, and competitive advantage) as well as the analysis of small and medium enterprises' activities in the context of economic crises, constitutes the study's theoretical originality. This study specifically sought to show that there is a noticeable difference in the link between strategic entrepreneurship and MSME performance during tumultuous times and stable economic conditions. By examining MSME strategic behavior in a sustainable context, for instance, it is possible to draw the conclusion that business owners should combine a number of different strategic stances to obtain the best results (Atuahene-Gima and Ko 2001; Bayiley and Behaylu 2022; Deutscher et al. 2016; Ho et al. 2016).

Given the period of time it was conducted and the characteristics of the sample, this study also presents a distinctive contribution regarding MSMEs in Cibinong District, Bogor Regency during the COVID-19 pandemic of 2019–2021. Due to management's cognitive biases in their view of companies' historical conduct, research on post-crisis business strategies runs the risk of being prejudiced and unreliable (Bao et al. 2011). The results are also applicable to all businesses that fit the selection criteria because the sample of domestic companies was representative.

The identified methods of managing MSMEs that ensure a firm performs at its best during the COVID-19 pandemic are of practical value to top managers, MSME decision-makers, and those in charge of creating and implementing strategies. It is crucial for MSME managers to understand that combining SE components—which improves performance in stable circumstances—can have detrimental effects during economic crises. In the latter scenario, they should concentrate on encouraging entrepreneurial behavior, which typically entails creativity, a readiness to take calculated risks when creating new goods and services, and the proactive pursuit of and use of new business prospects (Cho and Lee 2018; Soininen et al. 2012).

## 5. Conclusions and Implications

### 5.1. Conclusions

In this study, the performance of MSMEs in Cibinong District, Bogor Regency, was compared to the impact of strategic entrepreneurship strategies. Environmental factors, individual resources, resource orchestration, and competitive advantage were the four strategic entrepreneurship practices that could enhance the performance of small and medium-sized enterprises selected for study.

The results from the analyses showed that the input characteristics of environmental variables, organizational resources, and personal resources are in the 'excellent' category. This demonstrates that the implementation of strategic entrepreneurship in women-owned fish processing MSMEs in Bogor Regency has been proceeding well. The orchestration resource displayed an average ranking in the excellent category in the process dimension. In terms of output, the competitive edge, value creation, performance creation, and other benefits all had averages that fell into the excellent category. As a result, strategic entrepreneurship management in food-related MSMEs in Bogor Regency was already performing quite well.

The interaction of the strategic entrepreneurship variables demonstrated that the environment and individual resources had a favorable and significant impact on resource orchestration. Resource management had an impact on the development of competitive advantage. The production of performance and benefits for people, organizations, and society were also impacted by competitive advantage.

Based on the study's findings, it is determined that MSME management should strategically structure its resource portfolio, invest in human capital, and integrate by projecting both opportunity- and advantage-seeking behaviors to update its quality goods. Managers of MSMEs should be aware that entrepreneurial innovation and strategic resource management play a significant role in determining MSME performance since they foster creativity, introduce cutting-edge technology, and promote R&D activities that increase an organization's profitability.

### 5.2. Limitations and Future Research

Certain limitations should be considered when evaluating these findings. The first point is that the cross-sectional data used reflect recent business performance. The long-term effect of strategic entrepreneurship on the fish processing performance of women-owned MSMEs may be a topic for future studies. Second, the primary dependent variable in the study was managers' individual perceptions of the activities of the organizations, which was a subjective indication of such activities. Despite the approach's dependability, more research is needed to clarify the outcomes.

Third, we solely considered the direct impacts of strategic entrepreneurship components or their mixtures. The authors of future studies could decide to concentrate on other factors that moderate the relationship between strategic entrepreneurship and corporate performance. It may also be necessary to conduct replication studies employing a variety of samples, such as those made up of huge businesses and state-owned corporations.

### 5.3. Policy Implications

Fish processing in Bogor Regency is one of the most important contributors to the region's economic growth; thus, it is necessary for the government to maintain and generate the development of this business. Since environmental factors, individual resources, resource orchestration, and creating value and competitive advantages are essential factors in increasing the performance of micro, small, and medium fish processing enterprises in the Bogor Regency, the government should consider several policy recommendations related to these factors.

The first policy recommendation is to provide training for businesswomen to enhance product quality and produce differentiated products, including branding and packaging. The second policy is to promote the entrepreneurial mindset of businesswomen by providing capacity building and training. The third policy is to improve market access for their processed fish products by promoting these products at exhibitions and on government websites or social media. The last policy is to provide easy access to credit for businesswomen by providing credit terms and publishing credit recommendations. In addition, the local government should collaborate with banks, especially local banks, to provide low-interest credit.

**Author Contributions:** Conceptualization, A.A.Y.; methodology, A.A.Y., E.R. and R.; investigation, A.A.Y.; resources, A.A.Y.; writing—original draft preparation, A.A.Y.; writing—review, A.A.Y., E.R. and R.; and editing, E.R. and R. All authors have read and agreed to the published version of the manuscript.

**Funding:** This research received no external funding.

**Institutional Review Board Statement:** Not applicable.

**Informed Consent Statement:** Not applicable.

**Data Availability Statement:** Not applicable.

**Acknowledgments:** Directorate of Research and Community Service (DRPM) Universitas Padjadaran.

**Conflicts of Interest:** The authors declare no conflict of interest.

## Appendix A

**Table A1.** Survey (The original survey used with the participants was translated into this version).

| Latent Variable | Manifest Variable | Code | Value | | | | |
|---|---|---|---|---|---|---|---|
| | | | 1 | 2 | 3 | 4 | 5 |
| Environmental Factors | Consumer demand continues to increase | FL1 | | | | | |
| | Can adapt to business environment | FL2 | | | | | |
| | Can get capital access | FL3 | | | | | |
| | Can fulfill resource | FL4 | | | | | |
| | Production complexity | FL5 | | | | | |
| | Marketing complexity | FL6 | | | | | |
| Organizational Resources | Having new ideas and creativity with employees | SO1 | | | | | |
| | Give employees a chance to take risks | SO2 | | | | | |
| | Employee failure is tolerated | SO3 | | | | | |
| | Having programs for employees to learn | SO4 | | | | | |
| | Supporting innovation | SO5 | | | | | |
| | Continuous change | SO6 | | | | | |
| | Commitment to improvement | SO7 | | | | | |
| | Protect innovation threats | SO8 | | | | | |
| | Delivering opportunities | SO9 | | | | | |
| | Reasonable entrepreneurship | SO10 | | | | | |
| | Revisit entrepreneurship principles | SO11 | | | | | |
| | Link strategic management and entrepreneurship | SO12 | | | | | |
| Individual Resources | Introducing opportunity | SI1 | | | | | |
| | Entrepreneurial agility | SI2 | | | | | |
| | Real logical thinking | SI3 | | | | | |
| | Framework of entrepreneurship | SI4 | | | | | |
| | Opportunity collection | SI5 | | | | | |
| Resource Orchestration | Structuring resource portfolio | OS1 | | | | | |
| | Bundling resource | OS2 | | | | | |
| | Leverage capability | OS3 | | | | | |
| Creating value and competitive advantage | Customer relationship | NS1 | | | | | |
| | Different in service | NS2 | | | | | |
| | Business cost | NS3 | | | | | |
| | Differentiation | NS4 | | | | | |
| | Focus | NS5 | | | | | |
| Creating performance | Able to tax payment | KM1 | | | | | |
| | Improvement in assets | KM2 | | | | | |
| | Creating new jobs | KM3 | | | | | |
| | Improvement in selling | KM4 | | | | | |

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
