# Peer review of "Strategic Entrepreneurship and the Performance of Women-Owned Fish Processing Units in Cibinong District, Bogor Regency"

_economies, doi:10.3390/economies11030088_

Round 1

Reviewer 1 Report

Dear authors,

Thank you for sending me the article. Please find comments below!

The introduction section should present the significant (practical and theoretical) issues to be studied in the topics and case study presented. The important of conducting research on the issues would be the motivation for the research. If they are not investigated, what are negative impacts will raise as consequence.

Hypotheses should be written in active sentences.

Table 1 and Table 2 are sufficiently explain in good paragraphs. No need the tables.

The manifest variables are the dimensions of the latent ones. References that explain this relationship are needed to explain measurement of the latent variables.

An explanation of the survey to obtain the primary data should be with the types of questions in the questionnaire, the number of target respondents, ethics in compiling the questionnaire and collecting data.

A Common Method Bias (CMB) test before analyzing data is needed to clarify the data with no significant bias.

Good luck!

Author Response

Dear Prof

Thank you very much for your attention, I've attached files with to response  your comments.

I sincerely appreciate it. You have made very wise suggestions. I believe that your suggestions have improved our article. 

Sincerely,

Author

Reviewer 2 Report

Dear Authors,

find my comments attached. I think the introduction part should be stronger. Also, the methodology is lacking some important information. The results (H rejection and acceptation) should be highlighted!

All the best!

Author Response

(The authors gave the same response as above.)

Round 2

Reviewer 1 Report

Thank you very much for sending me the revised version.

Some words that are confusing in the introduction section I write below.

"their" in line 31, "this" in line 44, "new" in line 49, "all sizes businesses" in line 55. "small and large" in line 59. "Entrepreneurship, including Indonesia" in line 60.

So, proofreading before resubmitting is essential for this article.

After reading the article carefully, I have not seen the theories as foundations of research framework provided. What theories can be used to explain the importance of strategic entrepreneurship in SMEs?

Again, the reasons not to use a Common Method Bias test before the data is analyzed is very important!

After reading sections 2 to 5, I highly recommend that this article improve language style extensively.

Good luck!

Author Response

Dear Prof

Thank you very much for your attention, I've attached files to response  your comments.

I sincerely appreciate it. You have made very wise suggestions. I believe that your suggestions have improved our article. 

Sincerely,

Author

Round 3

Reviewer 1 Report

Dear authors,

I can see the efforts that have gone into the revision process. I highly
recommend proofreading this article before it is accepted or published
in the journal. I recommend one related reference below, you can add
them in the Introduction part.
https://www.mdpi.com/2227-7099/10/7/162

Good luck!